# Nerve Growth Factor and the Role of Inflammation in Tumor Development

Giampiero Ferraguti [1,†], Sergio Terracina [1,†], Luigi Tarani [2], Francesca Fanfarillo [1], Sara Allushi [1], Brunella Caronti [3], Paola Tirassa [4], Antonella Polimeni [5], Marco Lucarelli [1,6], Luca Cavalcanti [7], Antonio Greco [7] and Marco Fiore [4,*]

1   Department of Experimental Medicine, Sapienza University of Rome, 00185 Rome, Italy
2   Department of Maternal Infantile and Urological Sciences, Sapienza University of Rome, 00185 Rome, Italy
3   Department of Human Neurosciences, Sapienza University Hospital of Rome, 00185 Rome, Italy
4   Institute of Biochemistry and Cell Biology (IBBC-CNR), Department of Sensory Organs, Sapienza University of Rome, 00185 Rome, Italy
5   Department of Odontostomatological and Maxillofacial Sciences, Sapienza University of Rome, 00185 Rome, Italy
6   Pasteur Institute, Cenci Bolognetti Foundation, Sapienza University of Rome, 00185 Rome, Italy
7   Department of Sensory Organs, Sapienza University of Rome, 00185 Rome, Italy
*   Correspondence: marco.fiore@cnr.it
†   These authors contributed equally to this work.

**Abstract:** Nerve growth factor (NGF) plays a dual role both in inflammatory states and cancer, acting both as a pro-inflammatory and oncogenic factor and as an anti-inflammatory and pro-apoptotic mediator in a context-dependent way based on the signaling networks and its interaction with diverse cellular components within the microenvironment. This report aims to provide a summary and subsequent review of the literature on the role of NGF in regulating the inflammatory microenvironment and tumor cell growth, survival, and death. The role of NGF in inflammation and tumorigenesis as a component of the inflammatory system, its interaction with the various components of the respective microenvironments, its ability to cause epigenetic changes, and its role in the treatment of cancer have been highlighted in this paper.

**Keywords:** apoptosis; cancer stem cells; metastasis; epigenetic; inflammation; microenvironment; neurotrophins; NGF; tumor

## 1. Introduction

Inflammatory processes play a multifaceted role in tumor development and progression [1–7]. While the immune system's inflammatory response is typically a defense mechanism against infections and tissue damage, chronic or persistent inflammation can contribute to the initiation, growth, and spread of certain types of tumors. Thus, the understanding of the relationship between inflammation and cancer could better focus on anti-inflammatory therapies and drugs that target inflammatory pathways as potential strategies for cancer prevention and treatment [8]. In recent years, a growing body of evidence has emphasized the involvement of neurotrophins (NTs) in the complex landscape of both inflammation and cancer biology, showing their significant role in determining tumor cell growth and survival, particularly in certain types of cancers expressing NT receptors on their cell surfaces [9,10]. One of the most well-studied NTs is the nerve growth factor (NGF), which binds to its specific receptor, tropomyosin-related kinase A (TrkA), expressed on various types of cancer cells, including those derived from the brain, prostate, breast, and pancreas, among others [11–13]. NGF contributes to inflammation by acting as a signaling molecule that stimulates immune cells to release cytokines and enhances the sensitivity of sensory nerves, contributing to the perception of pain and hypersensitivity in inflamed

tissues [14]. NTs promote tumor growth by stimulating cancer cell proliferation and suppressing apoptosis, which physiologically helps eliminate damaged or unwanted cells from the body [15]. Furthermore, NTs can stimulate the production of pro-angiogenic factors, modulate the tumor microenvironment (TME), and induce the epithelial–mesenchymal transition (EMT), leading to increased cell mobility and invasiveness [16].

Overall, the activation of NGF signaling pathways in cancer cells can contribute to aggressive tumor growth, metastasis, and therapy resistance. Originally identified for its pivotal role in neuronal development and function, NGF has emerged as a critical player in the growth and survival of various tumor types. The intriguing aspect of NGF's contribution to cancer lies in its dual role, acting as both an oncogenic factor, fueling tumor cell growth, and as a pro-apoptotic mediator, promoting tumor cell death under certain circumstances [17–19]. The context-dependent actions of NGF in cancer underscore the complexity of the signaling networks in which it operates and its interaction with diverse cellular components within the TME. Overall, NGF's involvement in both inflammation and tumors further highlights its complex role in regulating physiological processes and disease states. Understanding the intricate interactions of NGF with various cell types and pathways is crucial for developing targeted therapies that can modulate its effects in different disease contexts, offering a potential therapeutic strategy to inhibit tumor progression and improve patient outcomes. However, further research is needed to fully understand the complexities of NGF signaling in different cancer types and to develop effective and safe targeted therapies.

This narrative report aims to provide a summary and subsequent review of literature evidence on the role of NGF and its role in regulating tumor cell growth and death as an inflammatory factor.

## 2. NGF and Inflammation and Tumor Growth

### 2.1. Nerve Growth Factor and Neurotrophins

NGF is a crucial neurotrophic factor responsible for the growth, survival, and maintenance of neuronal and non-neuronal cells [20,21]. Discovered in the 1950s, it was one of the first NTs identified and extensively studied for its role in the development and function of the nervous system [22,23]. NGF belongs to the family of NTs, which includes other proteins like Brain-Derived Neurotrophic Factor (BDNF), Neurotrophin-3 (NT-3), and Neurotrophin-4/5 (NT-4/5) [24,25]. NGF is synthesized as a precursor, called pro-NGF, which after processing generates the mature NGF molecule. NGF is primarily produced by various cell types, including immune cells, endothelial cells, and tissues within the nervous system [14,26–28].

Pro-NGF has a physiological role that goes beyond that of a simple precursor, especially in the nervous system, where it may possess pro-apoptotic activity [29]. Indeed, pro-NGF regulates apoptosis and inflammation and is associated with several neurodegenerative diseases, myocardial infarction, and diabetes [30–34]. The actions of NGF are mediated through its interaction with specific receptors. There are two primary receptors associated with NGF:

(1) TrkA is the high-affinity receptor for NGF; its activation triggers a cascade of intracellular signaling events, including the MAPK/ERK pathway and the PI3K/Akt pathway [35,36]. These pathways play essential roles in cell growth, differentiation, and survival. TrkA is expressed on the surface of neurons and other cell types, enabling NGF to exert its neurotrophic effects;

(2) p75 neurotrophin receptor (p75NTR) has a lower affinity for NGF but plays a modulatory role in NGF signaling [20,37]. It can interact with TrkA and enhance its binding affinity for NGF, influencing the cellular responses elicited by NGF. p75NTR is mostly involved in processes like cell death, survival decisions, and axonal growth.

Furthermore, NGF may attach to the membrane receptor sortilin, which has been demonstrated to participate in cancer growth [38], and also to neuropilin-1 (NRP1), a nociceptor-enriched co-receptor for NGF that is necessary for the TrkA signaling of pain [39–41].

Indeed, modifications in NGF levels in the serum and plasma have been shown during the beginning and evolution of many health conditions, including the post-partum period, stressful events, cardiometabolic disruptions, aging, alcohol addiction, and other pathophysiological conditions, such as psychiatric, neurological, and immune disorders [42–52]. Thus, the understanding of the interplay between NGF and its receptors is crucial not only for comprehending the complexities of neuronal development and function but also for exploring potential therapeutic interventions in various diseases, including neurodegenerative disorders and certain types of cancers where NGF signaling may be implicated [51,53–55].

### 2.2. Inflammation

Inflammation is a natural response by the body's immune system to protect against harmful stimuli like pathogens, irritants, or damaged cells. It is a complex biological process that aims to eliminate the initial cause of cell injury, clear out damaged cells, and initiate tissue repair [56]. Actually, in medicine, inflammation, a term coined by the Romans, lacks a precise, universally accepted definition, varying in interpretation based on context and individual perspectives [57]. It often carries a negative connotation as an uncontrolled reaction likened to a destructive wildfire, requiring immediate containment.

However, overshadowed in this view is the fundamental role that inflammation plays in both maintaining health and ensuring survival. Inflammation involves a complex interplay of cells, chemicals, and molecular signals including blood vessel dilatation (causing redness and heat) and increased permeability, allowing immune cells and fluids to move from the bloodstream into the tissues (leading to swelling); the cellular release of chemicals such as histamine, cytokines, and prostaglandins, which help to trigger the immune response and promote healing; and finally the activation of immune cells which migrate to the affected area to destroy pathogens or damaged cells [58].

Cytokines play a pivotal role in orchestrating or modulating inflammation, confirming not only the presence and magnitude of inflammation but also guiding treatment decisions. Inflammation can be distinguished as acute and chronic. Acute inflammation is the body's immediate and short-term response to an injury or infection [59]. It is characterized by symptoms like redness, swelling, heat, pain, and sometimes loss of function [60,61]. Chronic inflammation, on the other hand, is long-term and can last for weeks, months, or even years [62]. It occurs when the immune system's response persists, often due to underlying health conditions, such as autoimmune disorders, ongoing infections, obesity, or prolonged exposure to irritants like smoke.

Lifestyle factors like stress, poor diet, lack of exercise, and environmental toxins can contribute to chronic inflammation [63]. While inflammation is a vital part of the body's defense mechanism, chronic inflammation can be a problematic event leading to tissue damage and various diseases including diabetes, cancer, cardiovascular diseases, eye disorders, arthritis, obesity, autoimmune diseases, and inflammatory bowel disease [64].

Interestingly, inflammation is at the base of various conditions and a plethora of etiopathogenetic events; for example, hepatitis (liver inflammation) has been associated with viral infections, excessive alcohol consumption, certain medications, or autoimmune responses [62,65–68]. Long-term inflammation of the liver can lead to cirrhosis and liver cancer [69,70]. Inflammatory states and diseases cover a wide range of conditions affecting various parts of the body, varying in severity and requiring different treatments, including medications, lifestyle changes, and sometimes, in more severe cases, surgical interventions or specialized therapies.

The widespread use of anti-inflammatory medications assumed to counteract all inflammatory responses potentially may hinder the body's ability to fully recover [71,72]. Indeed, not all situations warrant an inflammatory response (such as blunt trauma and exposure to toxins), but since inflammation affects both unhealthy and healthy tissues without discrimination, it should be treated when it has the potential to persist or spread uncontrollably, causing prolonged damage. Effective management often entangles a multi-

disciplinary approach that requires the support of specialized healthcare professionals in this specific condition. Actually, it has been suggested that an effective way to guide therapy for inflammation is to assess a combination of markers associated with inflammation and fibrosis, such as C-reactive protein, ferritin, serum amyloid A (SAA), pro-calcitonin, and transforming growth factor-β (TGF-β, a significant contributor to fibrosis), alongside cytokine profiling [73].

### 2.3. Role of NGF in Carcinogenesis

NGF plays a significant role in various aspects of human health, including its involvement in tumors. Overall, evidence indicates that NGF is unable to generate cell carcinogenesis alone, both in normal neuronal and non-neuronal cells/tissues; however, it could be a major determinant in the case of co-expression with pro-carcinogenic molecules [74]. Quite intriguingly, NGF was initially discovered by R. Levi-Montalcini nearly 60 years ago in the context of a transplantation experiment involving a malignant mouse sarcoma [75,76].

#### NGF as a Tumor Growth Facilitator or Suppressor

Depending on the tumor's origin, pro-survival signaling can be facilitated through TrkA and/or p75NTR receptors [77]. In breast cancer, NGF plays a crucial role in stimulating proliferative signaling via TrkA and pro-survival signaling through p75NTR [35]. Furthermore, the activation of p75NTR in breast cancer promotes increased resistance to cell death induced by chemotherapeutic treatments. On the other hand, the role of p75NTR in prostate cells is distinct since p75NTR mediates cell death and acts as a tumor suppressor in the case of normal prostate cells [78]. In prostate cancer, the expression of p75NTR is lost, contributing to tumor progression, death evasion, uncontrolled proliferation, and metastasis to distant sites [79]. Interestingly, other mechanisms were found in recent studies. For example, NGF plays a significant role in liver cancer progression and metastasis, exerting wide influences on liver cancer cell polarity and motility by regulating signaling pathways involved in cell movement, cytoskeletal organization, and cellular polarity [80]. Heightened NGF disrupts cell polarity, boosts cell movement, triggers changes related to cell transition, rearranges the cell's structural framework, and protects cells from apoptosis and detachment-induced cell death [80]. In Table 1, we report the main role of NGF and its receptors in various cancers.

**Table 1.** Detailed overview of the role of NGF and its receptors in various types of tumors. While NGF's primary function is related to neural development and function, its relationship with tumors is complex and multifaceted. The involvement of NGF in tumors is not as straightforward as in normal nerve growth, and its effects on different types of tumors can vary. Acetylcholine, Ach; A disintegrin and metalloprotease 17, ADAM17; protein kinase B, Akt; extracellular signal-regulated kinase, ERK; F-box-only protein 22, FBOXO22; hypoxia-inducible factor 1 subunit alpha, HIF1α; nerve growth factor, NGF; non-small-cell lung cancer, NSCLC; neurotrophic tyrosine receptor kinase, NTRK; nuclear factor kB, NF-kB; p75 pan-neurotrophin receptor, p75NTR; programmed death-ligand 1, PD-L1; rearranged during transfection, RET; small nuclear ribonucleoprotein polypeptide A, SNRPA; tissue inhibitor of metalloproteinases, TIMP; tropomyosin receptor kinase, Trk; vascular endothelial growth factor, VEGF. (*) MYCN is amplified in 20% of neuroblastomas and correlates with aggressive phenotype and poor prognosis. (**) Perineural invasion driven by the TME has been identified as a key pattern of several malignancies including breast, pancreatic, and prostate cancers.

| Kind of Tumor | Role of NGF | References |
|---|---|---|
| **Brain tumors** | • Pro-NGF, NGF, and TrkA are expressed in numerous brain tumor cells (especially glioblastoma) and can promote cell survival and growth.<br>• p75NTR proteolysis is required for brain tumor proliferation.<br>• NGF concentrated in centrosome can phosphorylate TrkA, which may in return phosphorylate tubulin and promote mitotic spindle assembly, causing mitogenic effects in glioma cells.<br>• In ependymoblastoma tumors, NGF exerts a marked action on differentiation rather than proliferation in vitro.<br>• The potential of pro-NGF, NGF, and its receptors as clinical biomarkers and therapeutic targets has been highlighted. | [81–89] |

**Table 1.** *Cont.*

| Kind of Tumor | Role of NGF | References |
|---|---|---|
| Breast Cancer | <ul><li>NGF is both synthesized and released by breast cancer cells.</li><li>NGF exerts mitogenic, antiapoptotic, and angiogenic influences on breast cancer cells by engaging distinct signaling pathways that encompass the participation of TrkA and NGFR/p75NTR receptors.</li><li>Pro-NGF signaling has been linked to breast cancer invasion and metastasis.</li><li>NGF and its receptors have been identified as diagnostic and prognostic tools.</li><li>NGF and its receptors are promising therapeutic targets for breast cancer.</li><li>Norepinephrine/β2-Adrenergic Receptor pathway promotes cell proliferation and NGF production in triple-negative breast cancer.</li><li>Neurotoxin inhibition of sympathetic neural signaling in mammary tumors using 6-hydroxydopamine or genetic deletion of NGF or β2-adrenoceptor in triple-negative breast cancer tumor cells enhances the therapeutic effect of anthracycline chemotherapy by reducing metastasis in xenograft mouse models.</li><li>Electroacupuncture promotes apoptosis and inhibits axonogenesis by activating the p75NTR receptor for triple-negative breast xenograft in mice.</li><li>In rat models, it has been found that NGF from breast cancer may mediate spinal bone pain from metastasis via axonal growth and up-regulation of pain-associated neuropeptides.</li></ul> | [26,90–99] |
| Colorectal Cancer | <ul><li>NGF can stimulate the growth and survival of colon cancer cells and influence their invasive behavior.</li><li>NGF has been identified as a potential therapeutic target for the treatment of colon cancer.</li><li>NGF receptors may play a key role in androgen's effect on hormone-sensitive tumor cells.</li><li>NGF has been included in new immunogenomic prognostic risk scores for colorectal cancer.</li></ul> | [17,47,100–103] |
| Gastric Cancer | <ul><li>In gastric cancer, NGF and Trk receptor mRNA expression are down-regulated.</li><li>NGF Trk receptors may elicit cell apoptosis by a Ras or Raf signal transduction pathway.</li><li>SNRPA (small nuclear ribonucleoprotein polypeptide A) enhances tumor cell growth in gastric cancer through modulating NGF expression.</li><li>ACh-NGF positive feedback loop may be the basis for the abnormal innervation observed in the TME and acts through the Trk receptors.</li><li>p75NTR inhibits the invasive and metastatic abilities of gastric cancer cells by down-regulating uPA and matrix metalloproteinase 9 proteins and up-regulating TIMP1 protein via the NF-kB signal transduction pathway.</li><li>p75NTR may be used as a new potential therapeutic target in metastatic gastric cancer.</li><li>Individual and co-expression patterns of NGF and heme oxygenase-1 may predict shorter survival of gastric carcinoma patients.</li></ul> | [104–108] |
| Head and Neck Cancer | <ul><li>NGF can stimulate morphological differentiation, adhesion, proliferation, and migration in head and neck cancer.</li><li>NGF controls cancer cell migration through Akt phosphorylation, suggesting a possible therapeutic role of Akt inhibitors.</li><li>NGF and TrkA are highly expressed in cases of head and neck carcinomas with and without perineural invasion and are associated with improved tumor cell survival. (**)</li><li>p75NTR and TrkC receptors demonstrate a different immunoreactivity profile in comparison to TrkA and TrkB receptors in the normal human pituitary gland and adenomas.</li><li>p75NTR and pattern of invasion predict poor prognosis in oral squamous cell carcinoma.</li><li>Pro-NGF may be a potential diagnostic biomarker for thyroid cancer.</li><li>In thyroid cancer, increased expression of the TrkA receptor has been correlated with tumor progression and lymph node invasion.</li></ul> | [10,44,109–117] |
| Leukemias | <ul><li>NGF induces the ERK signaling pathway through TrkA receptors stimulating the production and survival of immune cells, but the actual impact on hematological diseases is still to be determined.</li></ul> | [118–121] |
| Liver Cancer | <ul><li>NGF and TrkA are highly expressed in hepatocarcinoma tissue (especially in males).</li><li>p75NTR may provide a mechanism for selective apoptosis of hepatic stellate cells.</li><li>NGF and its receptors may play a role in cellular interactions involving hepatocarcinoma cells, hepatic stellate cells, arterial cells, and nerve cells in cancer tissues.</li><li>NGF regulates liver cancer cell polarity and motility associated with the invasion and metastasis process.</li><li>Various therapies for liver cancer have been found to act on NGF pathways or influence the pro-NGF/NGF balance.</li><li>NGF level evaluation may be useful to predict hepatic dysfunction after irradiation and has been studied as a possible biomarker for liver cancer.</li></ul> | [80,122–133] |

**Table 1.** *Cont.*

| Kind of Tumor | Role of NGF | References |
|---|---|---|
| **Lung Cancer** | • NGF has been linked to lung cancer progression, promoting the growth and survival of lung cancer cells and contributing to the development of chemoresistance.<br>• TrkA is increased in squamous cell carcinoma, NGF and pro-NGF are increased in both squamous cell carcinoma and in adenocarcinoma, and p75NTR is increased across all lung cancer histological subtypes compared to normal lung.<br>• NGF might play a role in the interaction between lung cancer cells and nerve fibers, leading to increased tumor growth.<br>• NGF and chemokines secreted by apoptotic astrocytes cause the formation of an inflammatory and immunosuppressive microenvironment, enabling the formation of a pre-metastatic niche in lung cancer brain metastases.<br>• Evidence of Trk fusion in NSCLC suggests the potential of NGF receptors as targets for further therapeutical applications. | [83,134–139] |
| **Ovarian Cancer** | • Ovarian cancer is marked by elevated levels of NGF and TrkA.<br>• NGF is involved in perineural invasion. (**)<br>• Through its interaction with the TrkA receptor, NGF decreases transcription of miR-145 levels, causing an increase in oncogenic proteins involved in promoting angiogenesis and cell proliferation and migration, as well as inhibiting apoptosis and influencing various molecules such as cyclooxygenase-2, ADAM17, and calreticulin, all essential for ovarian cancer progression.<br>• MicroRNAs may be associated with NGF/TrkA activation and alter key protein levels.<br>• The onset condition of ovarian cancer can be diagnosed through the detection of high or low expression of NGF and its receptors.<br>• TrkA may be considered a new potential tumor marker.<br>• NGF is also associated with increased resistance to chemotherapy in ovarian cancer cells.<br>• Blocking neurotrophin action could be a therapeutic target in treating ovarian cancer.<br>• Metformin treatment decreases the expression of c-MYC, β-catenin, and VEGF induced by NGF/TrkA while increasing oncosuppressor miRs, such as miR-145 and miR-23b. | [140–149] |
| **Neuroblastoma** | • NGF promotes the survival and growth of neuroblastoma cells by binding to TrkA on the surface of cancer cells.<br>• TrkA and TrkC are overexpressed in biologically favorable neuroblastomas: their expression was associated with an absence of N-myc amplification, lower disease stage, lower patient age, differentiated tumors, and a greater likelihood of spontaneous regression or responding well to therapy.<br>• TrkB is mainly expressed in unfavorable, aggressive neuroblastomas.<br>• p75 plays an important role in enhancing both the sensitivity of Trk receptors to low levels of ligand, as well as enhancing ligand-induced differentiation in TrkA/p75 but not TrkB/p75 cells.<br>• Targeting Trk receptors is a potential therapeutic strategy for neuroblastoma treatment.<br>• MYCN (*) targets estrogen receptor alpha (Erα) and thereby NGF signaling to maintain an undifferentiated and aggressive phenotype.<br>• RET and TrkA physically interact and can induce reciprocal activation in response to ligand activation. | [24,150–156] |
| **Pancreatic Cancer** | • NGF exerts both stimulatory and inhibitory effects on pancreatic cancers with the effect based on the expression levels and the ratio of TrkA and p75NTR.<br>• NGF from pancreatic stellate cells induces pancreatic cancer proliferation and invasion by the phosphatidylinositol 3-kinase (PI3K)/protein kinase B (AKT)/glycogen synthase kinase (GSK) signal pathway.<br>• TrkA expression in pancreatic cancer is a marker of tumor aggressiveness.<br>• Elevated p75NGFR expression is associated with a favorable prognosis.<br>• NGF overexpression combined with TrkA may contribute to perineural invasion by activating the Warburg effect, promoting tumor-derived exosomal miRNA-21 expression, prompting the hyperplasia of nerves, and inhibiting tumor cell apoptosis. (**)<br>• In pancreatic cancer, the high-glucose microenvironment promotes the invasion ability and raises the expression of NGF by upregulating HIF1α.<br>• NGF has been successfully used for diagnostic, therapeutic, and prognostic purposes in pancreatic cancer.<br>• Atorvastatin may exert an anti-tumor effect in pancreatic cells via the inhibition of NGF and other neurotrophin signaling pathways.<br>• In a mouse model, anti-NGF treatment beginning at 4 weeks may increase inflammation and negatively impact disease, while treatment starting at 8 weeks (after disease onset) reduces neural inflammation, neural invasion, and metastasis. | [13,15,16,157–167] |
| **Pediatric tumors** | • NGF Trk receptors play a key role in pediatric tumors, especially in brain cancers.<br>• Trk receptors play a key role in transducing the mitogenic effects of NGF and Trk inhibitors have been proven to be an important therapeutic tool for the treatment of NTRK fusion cancers.<br>• NGF administration may be an effective and safe adjunct therapy in children with optic atrophy due to optic gliomas. | [28,36,82,168–170] |

**Table 1.** *Cont.*

| Kind of Tumor | Role of NGF | References |
|---|---|---|
| **Prostate Cancer** | • NGF and its receptors are found in prostate cancer cells.<br>• NGF is released by both epithelial pancreatic cells and cancer-associated fibroblasts.<br>• NGF may be able to suppress tumor growth via an indirect effect, probably innervation or maturation of the tumor neo-vasculature.<br>• Aberrations and/or derangement of NGF signaling contribute to tumor growth and progression, enhancing the invasive potential of prostate cancer cells and promoting the development of cancer cells' neuroendocrine features.<br>• Expression of p75NTR reduces NGF-induced cell growth by activation of programmed cell death.<br>• Cross-talk between androgen receptors and NGF receptors in prostate cancer cells may have implications for a new therapeutic approach.<br>• Pro-NGF correlates with the Gleason score and is a potential driver of nerve infiltration in prostate cancer.<br>• Cancer-associated fibroblasts can activate Yes-associated protein (YAP1)/TEA domain (TEAD1) signaling and increase the secretion of NGF, therefore promoting perineural invasion. (**)<br>• FBXO22 mediates the NGF/TrkA signaling pathway and stimulates macrophage M2 polarization in prostate cancer bone metastases, promoting cell activity and osteogenic lesions. | [77,171–176] |
| **Skin tumors** | • NGF and its receptor seem to play a role in most skin tumors including malignant melanoma.<br>• In basal cell carcinoma and cutaneous squamous cell carcinoma, increased levels of NGF and TrkA, B, and C may reflect unique survival pathways.<br>• P75NTR may play a mechanistic role in invasive melanomas demonstrating perineural invasion. (**)<br>• Higher levels of Trk receptors in cutaneous squamous cell carcinoma cells may predict perineural invasion.<br>• Increased p75NTR expression in cutaneous squamous cell carcinoma perineurally may allow p75NTR immunohistochemical staining to be used for detecting sites of perineural invasion.<br>• NGF and TrkA are overexpressed in cervical squamous cell carcinoma.<br>• PD-L1 and NGF are co-expressed on spindle cells in the microenvironment of Merkel carcinoma, and TrkA receptors seem to play a major role.<br>• P75NTR has been identified as a useful marker to distinguish spindle cell melanoma from other spindle cell neoplasms of sun-damaged skin. | [54,177–186] |

While the primary role of NGF is related to the development and function of nerve cells, it also plays a main role in inflammation. Inflammation, as stated before, is a complex biological response triggered by the body's immune system to protect against harmful stimuli, such as tissue damage and pathogens. In this scenario, NGF can regulate the innervation and neuronal activity of peripheral neurons, inducing the release of immune-active cytokines, neuropeptides, and neurotransmitters [14,187–189]. Furthermore, NGF can also directly influence innate and adaptive immune responses through its interaction with various cells involved in the immune response, including mast cells, lymphocytes, and macrophages [28,190]. Actually, NGF has a variety of effects that can be either pro-inflammatory or anti-inflammatory depending on the expression of its receptors, which are dynamically regulated in immune cells depending on their state of differentiation and functional activity [191,192]. The seeming ambiguity is mainly due to the role of NGF as an endogenous molecule capable of triggering immune responses while also initiating pathways that control inflammation and prevent excessive tissue damage so that altered expression of its receptors could hinder NGF's ability to engage the regulatory feedback processes for finally sustaining the perpetuation of inflammation in conditions such as chronic inflammatory diseases or autoimmune disorders [14]. Additionally, as NGF can contribute to the sensitivity and pain associated with the neurogenic inflammation of tissues, a potential role of NTs has been suggested as novel treatment strategies in chronic inflammatory diseases [193,194].

More specifically, NGF has intricate connections with neuroinflammation, which involves complex interactions between immune cells, glial cells, and various signaling molecules. Microglia act as the resident immune cells of the central nervous system (CNS) and can become activated in response to injury or inflammation [195]. NGF can modulate the activation and function of glial cells, particularly microglia and astrocytes, influencing their release of inflammatory mediators [196]. It can both promote and dampen the release

of various cytokines depending on the context, contributing to the fine-tuning of the inflammatory response in the CNS [157,197,198] (see Figure 1).

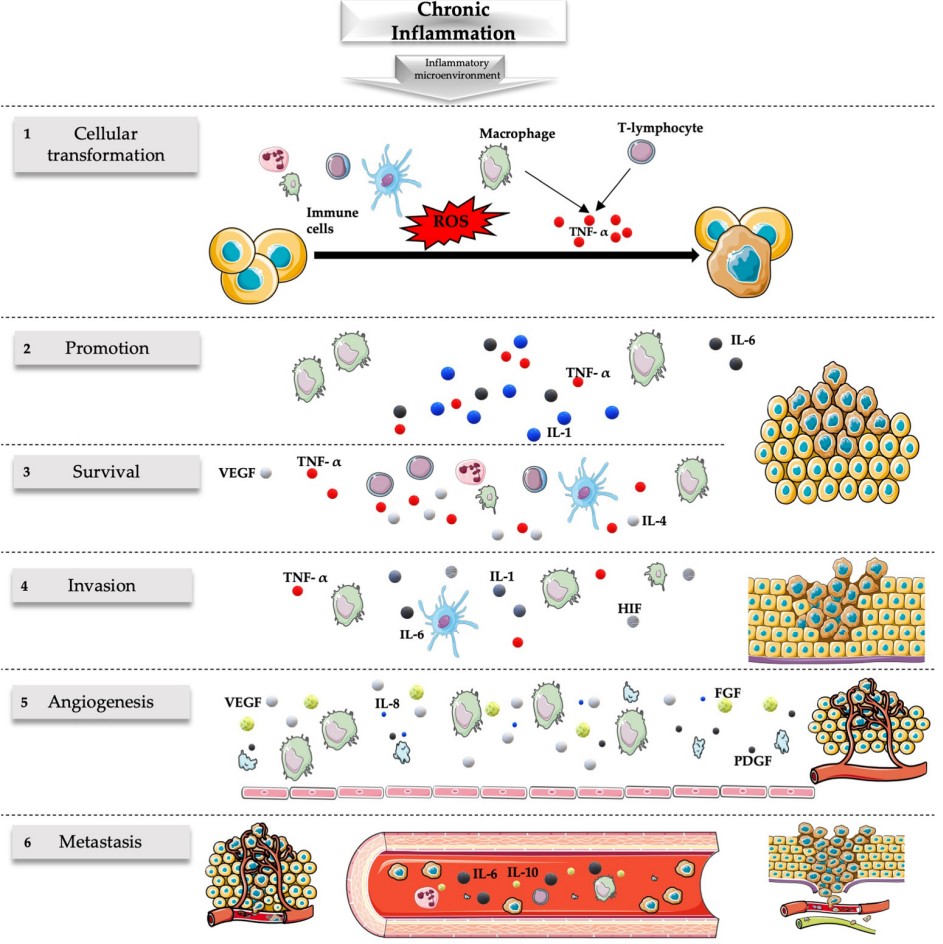

**Figure 1.** Role of chronic inflammation in cancer. Chronic inflammation originating from persistent stimuli has been associated with many steps of the carcinogenesis process including transformation, promotion, survival, proliferation, invasion, angiogenesis, and metastasis. (1) Cellular transformation is favored by the mutagenic action of ROS released by the immune cells and the action of TNF-α released by macrophages and T lymphocytes. (2) Carcinogenesis is promoted by various cytokines released during the chronic inflammatory process, including IL-1, IL-6, and TNF-α released by macrophages. (3) The survival of the tumor is associated with an ineffective response of the immune system associated with the defective action of inflammatory cells (like the release of IL-4 and IL-5 by T cells associated with T-helper 2 but not T-helper 1 responses) related to the release of various molecules like TNF-α, VEGF, Fas ligand, and transforming growth factor-β. (4) The invasion is favored by numerous molecules during inflammatory states; some of the most important are those associated with hypoxia including HIF, TNF-α, IL-1, and IL-6. (5) Inflammatory cells, especially macrophages, but also endothelial cells and platelets stimulate vascular growth through the release of many angiogenic factors (e.g., VEGF, IL-8, FGF, PDGF), sustaining the nutritional needs of the tumor and favoring its survival and migration. (6) Numerous cytokines and factors released during inflammation can lead to metastatic events of tumors including IL-6 and IL-10. The inflammatory microenvironment has a major role in this setting with implications for the prevention and treatment of cancer. FGF, fibroblast growth factor; HIF, hypoxia-inducible factor; IL, interleukin; PDGF, platelet-derived growth factor; ROS, reactive oxygen species; VEGF, vascular endothelial growth factor; TNF-α, tumor necrosis factor. Parts of the figure were drawn by using pictures from Servier Medical Art. Servier Medical Art by Servier is licensed under a Creative Commons Attribution 3.0 Unported License (https://creativecommons.org/licenses/by/4.0/, accessed on 21 January 2024).

On the other hand, NGF exhibits neuroprotective effects by mitigating the harmful consequences of neuroinflammation and supporting neuronal survival and function, potentially counteracting the detrimental effects of excessive inflammation on neurons [199,200]. Furthermore, dysregulation of NGF signaling has been associated with various neurological disorders characterized by neuroinflammation, such as Alzheimer's disease, Parkinson's disease, multiple sclerosis, and neuropathic pain conditions [201,202]. In these disorders, altered NGF levels or signaling pathways contribute to the progression of neuroinflammation and neuronal damage, so it has been suggested that modulating NGF levels or its interactions within the CNS may offer potential avenues for managing neuroinflammatory conditions and related neurological disorders [203,204].

Unfortunately, inflammation plays a multifaceted role in tumor development and progression. While the immune system's inflammatory response is typically a defense mechanism against infections and tissue damage, chronic or persistent inflammation can contribute to the initiation, growth, and spread of certain types of tumors in several ways [69]. Cycles of tissue damage and subsequent repair processes can create an environment conducive to genetic mutations and abnormalities, increasing the likelihood of cancerous changes in cells [205]. Inflammation can indeed sustain tissue damage and attract immune cells to the site of tissue damage.

Hallmarks of cancer-associated inflammation include the presence of infiltrating leukocytes, cytokines, chemokines, growth factors, lipid messengers, and matrix-degrading enzymes [206].

Some of these immune cells, like certain types of macrophages and lymphocytes, can produce factors that support tumor growth and suppress the immune system's ability to eliminate cancer cells [207]. Inflammatory signals can stimulate angiogenesis for oxygen provision and nutrients to the tumor cells, aiding their proliferation and survival [10,208]. Managing chronic inflammation, either through lifestyle changes or medication, may play a role in reducing the risk of certain cancers or improving treatment outcomes [209].

## 3. NGF and the TME

The TME is a complex, highly heterogeneous, and dynamic ecosystem of cells, molecules, and structures that surround and interact with cancer cells within a tumor [210]. The TME is mainly constituted of cancer cells, stromal cells, immune cells, ECM, blood vessels, and signaling molecules [211]. It is often characterized by immune evasion, hypoxia, acidic pH, and stromal changes. The TME is significantly influenced by inflammation, creating a complex milieu that can either promote or inhibit tumor growth [212]. Chronic inflammation in the TME can create a supportive environment for tumor growth, survival, and metastasis by fostering angiogenesis, extracellular matrix (ECM) remodeling, and immune suppression [213]. Before metastasis occurs, the primary tumor can alter the TME within a particular tissue, leading to the initiation of a pre-metastatic niche (PMN) [214]. This phenomenon presents a valuable window of opportunity for the primary tumor cells to adapt, survive, and subsequently thrive, ultimately creating favorable conditions for colonization in the target organ during the subsequent metastatic process [215].

NGF can influence the TME, affecting the interactions between cancer cells and surrounding stromal cells, immune cells, and other components, contributing to both tumor growth and suppression [74]. On one hand, NGF can induce apoptosis in some cancer cells, limiting their survival, and may modulate immune responses within the TME, influencing the anti-tumor immune response [19]. One intriguing aspect of NGF's protective function is its capacity to maintain the physiological activity within the tissue microenvironment, thereby safeguarding tissue and organ functionality. This effect has been notably observed in the preservation of corneal nerve cells and the regulation of homeostasis across various systems, including the nervous, immune, and endocrine systems [216]. On the other hand, evidence suggests that NGF can promote endothelial cell proliferation and angiogenesis by stimulating pro-angiogenic factors, facilitating the delivery of oxygen and nutrients to the tumor cells, and promoting tumor growth and metastasis [217]. Some of the most

important pro-angiogenic factors activated by NGF in tumors include vascular endothelial growth factor (VEGF), which stimulates the growth and permeability of blood vessels, facilitating the supply of nutrients and oxygen to tumor cells; fibroblast growth factor (FGF), which plays a role in the proliferation and migration of endothelial cells, which form the inner lining of blood vessels; interleukin-8 (IL-8), a chemokine that can recruit and activate endothelial cells, promoting angiogenesis; and Matrix Metalloproteinases (MMPs), which are enzymes that degrade the ECM, allowing endothelial cells to migrate and form new blood vessels [140,218,219]. Furthermore, NGF can influence the stabilization and activation of hypoxia-inducible factor 1-alpha (HIF-1$\alpha$), a transcription factor that plays a key role in regulating the expression of genes involved in angiogenesis in response to low oxygen levels (hypoxia) [100,158].

Two conceptual models, known as the "immunosurveillance model" and the "danger model," present differing viewpoints regarding the interaction between the immune system and cancer cells [74,207]. The immunosurveillance model proposes that the immune system identifies tumor cells as foreign invaders and eradicates them. Conversely, the danger model suggests that immune cells do not recognize cancer cells as threats due to specific signaling deficits. In both cases, the composition of the TME emerges as a critical factor influencing these immune responses. The interaction between immune cells that infiltrate the tumor, those that reside within it, as well as epithelial and stromal cells, results in the release of a multitude of pro-inflammatory, anti-inflammatory, pro-angiogenic, and anti-angiogenic agents [220]. Significantly, NGF has been discovered to support the survival and functionality of diverse immune cell types, which encompass mast cells, lymphocytes, macrophages, granulocytes, T- and B-cell subtypes, natural killer cells, and eosinophils [52]. These immune cells not only exhibit responses to NGF but also can generate and release NGF themselves, implying a role for this molecule both in innate and adaptive immune responses [14].

In the context of the TME, increased levels of NGF may be released and engage with immune cells in autocrine or paracrine manners [221,222]. Increased NGF levels could contribute to the establishment of an immunosuppressive TME, thereby fostering resistance to immunotherapy. Tumors can contain nerve fibers that release NGF, which interacts with nerve endings present in the TME [223]. This crosstalk between nerves and the tumor can modulate immune responses [150]. In some cases, NGF may promote the expansion of regulatory T cells (Tregs), which can suppress the activity of cytotoxic T cells and dampen the anti-tumor immune response [224].

NGF can modulate the production of cytokines and chemokines in the TME, influencing the function of immune cells, creating an immunosuppressive milieu, and favoring tumor growth and immune escape. Furthermore, NGF may indirectly induce the expression of immune checkpoint molecules such as the programmed death-1 ligand (PD-L1), a co-inhibitory factor of the immune response, on tumor cells or immune cells within the TME, further inhibiting the activity of cytotoxic T cells [225]. NGF can influence the polarization of tumor-associated macrophages (TAMs) within the TME. TAMs can adopt different activation states, and some of these states can promote an immunosuppressive environment that hinders effective anti-tumor immunity [226]. Cancer cells utilize a mechanism involving the generation of reactive oxygen species (ROS) to activate c-Jun [227]. This activation prompts the secretion of NGF, which in turn primes nociceptive nerves to produce calcitonin gene-related peptide (CGRP). Subsequently, neurogenic CGRP induces a cellular self-protective process known as autophagy in cancer cells, achieved by disrupting the mTOR–Raptor interaction via Rap1 signaling. Interestingly, strategies aimed at starving cancer cells of nutrients by targeting glycolysis or angiogenesis can inadvertently exacerbate the reciprocal relationship between cancer cells and nociceptive nerves. To break this cycle and enhance therapeutic outcomes, it could be beneficial to intervene by inhibiting neurogenic CGRP.

NGF plays a central role in the intricate signaling network within Cancer Stem Cell (CSC) metabolism. CSCs are key elements of the TME, possessing the ability to self-renew

and playing a pivotal role in the expansion and chronicity of malignant tumors favoring host immune surveillance escape, the release of pro-tumorigenic chemokines by innate immune cells, and the development of more aggressive tumors [9]. Some studies have suggested that NGF can activate signaling pathways that are associated with stem cell characteristics, such as the phosphoinositide 3-kinase (PI3K)/protein kinase B (Akt) pathway [141]. Activation of these pathways can contribute to the self-renewal and survival of CSCs [228]. NGF can affect the behavior of various cells within the TME, including immune cells, fibroblasts, and endothelial cells within the tumor, which can create an environment that supports CSC survival and function. The expansion and dissemination of CSCs are intricately linked to their energy supply, primarily through the insulin-like growth factor (IGF-1) receptor pathway controlling glucose uptake by acting on the insulin receptor substrate 1 (IRS1)/PI3K/AKT pathway and the nuclear factor kappa B (NF-Kb)/Snail/zinc finger E-box-binding homeobox 1 (ZEB1) pathway (crucial for the EMT) [229–231]. So, the insulin/IGF pathway collaborates with signaling platforms associated with cell proliferation and pluripotency, fostering the transcription of EMT factors. Now, NGF steps into the spotlight. The EMT is a biological process during which epithelial cells transform, losing their characteristic features and adopting mesenchymal properties. This transition results in heightened cell mobility and invasiveness, ultimately facilitating the spread of cancer cells to nearby tissues and distant organs. Emerging evidence suggests that NGF plays a significant role in triggering the EMT in specific cancer cell types, further enhancing their invasive potential and metastatic capabilities [232,233]. Furthermore, NGF has been shown to regulate insulin signaling and combat insulin resistance by enhancing glucose uptake, particularly in degenerating neurons, while TrkA can transactivate the insulin receptor (IR) signaling pathway and the oncogenic fusion protein Trk-T1, originating from thyroid carcinoma, can trigger the formation of the IRS–Growth factor receptor-bound protein 2 (IRS-Grb2) complex with the related pro-oncogenic processes (transformation of fibroblasts and intestinal epithelial cells). NGF signaling pathways have been shown to induce changes in cell morphology and enhance the migratory and invasive properties of these cancer cells, contributing to the aggressive behavior of tumors [80]. CSCs are often resistant to conventional treatments like chemotherapy and radiation, and NGF-mediated signaling pathways may enhance their ability to withstand these therapies. Given the potential role of NGF in CSC behavior, it has been suggested as a potential therapeutic target so that modulating NGF signaling pathways could be explored as a strategy to target CSCs and improve the effectiveness of cancer treatments.

## 4. NGF Receptor Expression in Cancer

Both NGF high-affinity receptors (Trk) and the low-affinity p75NTR receptor and their cross-talk have been found to play significant roles in various types of cancer (Table 1).

The activation of Trk receptors is initiated by the autophosphorylation of tyrosine residues within their intracellular domains. This activation subsequently triggers multiple downstream signaling pathways involving various enzymes and adaptor proteins, including extracellularly regulated kinases (ERKs), phosphatidylinositol 3-kinase (PI3K), and phospholipase C (PLC). These pathways collectively govern cell proliferation, differentiation, and survival, impacting both neuronal and non-neuronal cell types. Conversely, signaling through p75NTR leads to the activation of a c-jun N-terminal kinase (JNK) cascade, which is contingent on the specific adaptor proteins bound to the receptor. The activation of p75NTR, via JNK, leads to the activation of p53 and the expression of pro-apoptotic genes such as Bcl-2. Additionally, when the p75NTR collaborates with tumor necrosis factor receptor-associated factor 6 (TRAF6), it promotes the activation of the nuclear factor-kappa B (NF-kB) signaling pathway, which has a pro-survival effect. The diversity in activation signaling pathways mediated by NT receptors reflects the complexity of their functions, both within and outside the central nervous system, as well as in various pathological conditions.

TrkA is one of the high-affinity receptors for NGF typically expressed in neuronal cells. However, in cancer, aberrant TrkA expression has been observed in various types of tumors, including ovarian cancer, pancreatic cancer, and neuroblastoma [91,164]. In some cases, overexpression or activation of TrkA has been linked to tumor growth, survival, and angiogenesis [165]. Furthermore, higher levels of NGF and TrkA have often been associated with tumor perineural invasion (PNI) [110]. TrkB and TrkC also interact with neurotrophins like NGF and aberrant expression has been reported in different cancers [112]. TrkB is associated with aggressive tumor behavior, metastasis, and resistance to chemotherapy in some cases [24]. The p75NTR is a low-affinity receptor for NGF that, unlike Trk receptors, does not have intrinsic kinase activity. It can interact with various NTs, including NGF. The above-mentioned Trk signaling transduction and activation might also proceed in tumor cells when *NTRK* gene changes occur in these cells. On the whole, some types of gene variations associated with *NTRK* are known. The most common one in tumors is the *NTRK* gene fusion. Indeed, when the 3′ part of *NTRK* fuses with the 5′ part of another gene, the equivalent products of expression could elicit a subsequent oncogenic signal and ligand-independent kinase activation [234,235].

Trk fusion is widely distributed in several cancer types. The most evident cases are congenital fibrosarcoma and mammary analog secretory carcinoma, with 100% NTRK fusion discovered in these two cancers [236–238]. Several studies demonstrated that NTRK gene fusions are influential factors in some acute myeloid leukemias [239,240]. Furthermore, NTRK fusion has also been shown in other tumors, such as colorectal cancer and lung adenocarcinoma [241].

Regardless of gene fusion, the NTRK copy upregulation in nerve cells could also elicit an atypical overexpression of Trk kinases. Indeed, it has been shown that TrkA and phospho-TrkA in MDA-MB-231 cells were distinctly potentiated compared to those in standard breast tissues [152]. It was also demonstrated that TrkA overexpression was highly associated with tumor metastasis and cell proliferation [242]. Although Trk mutations in a single point are relatively rare during cancer development, several reports debated that Trk mutations in a single amino acid were associated with drug resistance versus Trk kinase inhibitors [235,243].

In cancer, p75NTR expression has been reported in multiple tumor types, including breast cancer, prostate cancer, and glioma with a complex and context-dependent role [9,26,37,244]. It can have both pro-survival and pro-apoptotic effects, depending on the cellular context and the presence of other co-receptors or ligands [85,136]. In some cases, p75NTR signaling has been associated with promoting cancer cell migration, invasion, and resistance to therapy.

Understanding the expression and function of NGF receptors, including both Trk receptors and p75NTR, is crucial for unraveling their roles in cancer development and progression. Researchers are actively investigating these receptors as potential therapeutic targets for cancer treatment, and some clinical trials are exploring the use of drugs targeting Trk receptors to inhibit tumor growth in specific cancer types with aberrant NGF receptor expression.

## 5. Epigenetics

Epigenetics refers to heritable changes in gene expression that occur without alterations in the DNA sequence itself, including methylation and histone modifications [245]. Epigenetic mechanisms play a significant role in regulating the inflammatory response, and conversely, inflammatory signals can induce changes in epigenetic marks [246]. Dysregulation of epigenetic mechanisms can contribute to chronic inflammation by altering the expression of genes involved in regulating the inflammatory response, while this chronic inflammation, in turn, is associated with aging and various diseases, including autoimmune disorders and cancer [247–249]. Furthermore, epigenetic modifications are critical in controlling the differentiation of immune cells so that changes in epigenetic marks can affect the function and response of immune cells, influencing the inflammatory state. On

the other hand, epigenetic modifications play a pivotal role in tumor development and progression [10,250]. DNA methylation involves the addition of a methyl group to DNA, often occurring at specific sites called CpG islands [245]. Hypermethylation of promoter regions can silence tumor suppressor genes, while hypomethylation can activate oncogenes, both contributing to tumor formation. The chromatin, made of DNA and histones, can be modified through various mechanisms (e.g., acetylation, methylation, phosphorylation). ATP-dependent chromatin remodeling complexes alter the chromatin structure, impacting gene expression, while alterations in histone modifications can influence gene expression by changing how tightly DNA is packaged around histones, affecting the accessibility of genes to transcription machinery so that dysregulation of histone modifications can contribute to tumor initiation and progression [251]. MicroRNAs (miRNAs) and long non-coding RNAs (lncRNAs) are involved in post-transcriptional gene regulation and can influence various cellular processes, including proliferation, apoptosis, and differentiation, contributing to tumor development [252,253]. Epigenetic changes contribute to tumor heterogeneity, where different cells within a tumor have varying genetic and epigenetic profiles, affecting treatment responses. Studies suggest that NGF can regulate the expression of DNA methyltransferases (DNMTs), enzymes responsible for adding methyl groups to DNA [254,255]. Altered DNMT activity due to NGF signaling can impact gene expression patterns, influencing neuronal development and function. Actually, many inflammatory mediators like tumor necrosis factor alpha (TNF-$\alpha$) and interleukin-6 (IL-6) can influence the expression of DNMTs and alter chromatin's histone [256,257]. These findings have clinical significance; an example is nonsteroidal anti-inflammatory drugs (NSAIDs), which can participate in various signaling pathways [104]. A recent study demonstrated that ibuprofen epigenetically increases p75NTR expression by downregulating promoter methylation and upregulating m6A-RNA-methylation in SGC7901 and MKN45 cells [258].

NGF signaling also affects histone modifications, inducing changes in histone acetylation and methylation, altering chromatin structure and gene accessibility to modulate gene expression during neuronal differentiation and plasticity [259]. NGF signaling can regulate the expression of specific miRNAs, modulating the expression of genes involved in cell growth, differentiation, and survival. Dysregulation of NGF signaling and associated epigenetic changes have been implicated in neurological and tumoral disorders [51,120,201]. Drugs targeting epigenetic modifications (e.g., DNAMT inhibitors, histone deacetylase inhibitors) aim to reverse aberrant epigenetic changes and restore normal gene expression, showing promise in cancer treatment [260]. Understanding the interplay between NGF and epigenetics holds promise for developing novel therapeutic strategies targeting epigenetic mechanisms to modulate NGF signaling in various conditions. Epigenetic signatures associated with inflammation-related diseases could serve as biomarkers for diagnosis, prognosis, and the development of personalized treatments.

## 6. Therapeutic Potential of Neurotrophins and Their Receptors in Tumors

Many have suggested effective approaches to inhibit cancer via NGF inhibition, suggesting a promising therapeutic potential [160,161]. Studies on anti-NGF treatments in mouse models showed that NGF impacts tumor progression and metastasis in a temporally dependent manner [157].

Since the identification of tyrosine kinase receptors (TRKs) in the 1960s and their subsequent categorization into distinct families, their significant involvement in various facets of cellular biology has become increasingly evident [135]. Simultaneously, it has become clear that aberrations in TRKs can significantly influence a wide range of cellular processes, including tumor development, shaping their biological behavior from the early stages to advanced disease progression. The identification of mutations in TRKs has paved the way for the development of novel pharmaceuticals designed to inhibit these receptors and their downstream pathways, signifying a groundbreaking advancement in cancer therapy with the potential to impact mortality rates. Consequently, numerous inhibitors targeting TRK receptors are now routinely employed in the treatment of diverse human

malignancies. Moreover, ongoing molecular research is actively exploring new mutations and drug candidates for cancer therapy, aiming to enhance patient quality of life and overall survival.

As stated before, NGF in the TME plays a multifaceted role, impacting various aspects of the immune response and playing a role in drug resistance in various types of cancers, contributing to the growth of inherently resistant populations and influencing the response of initially drug-sensitive cells. Enhanced NGF-TrkA signaling can promote cell proliferation through the PI3K/Akt and MAPK/ERK pathways, and sometimes, it has been associated with pro-survival signaling through p75NTR, preventing apoptosis triggered by chemotherapy drugs and allowing inherently resistant cells to survive and proliferate despite treatment [19,117,261]. Furthermore, NGF and pro-NGF may induce a state of quiescence in cancer cells, leading to cells not actively dividing, which are less susceptible to drugs that target rapidly dividing cells [232]. On the other hand, prolonged exposure to chemotherapy can lead to alterations in the TME, inducing increased NGF secretion by neighboring cells or stromal cells, which in turn could facilitate adaptive resistance mechanisms in sensitive cells (activation of survival pathways, DNA repair mechanisms, epigenetic changes), allowing cells to cope with the stress induced by chemotherapy and survive its effects [9,180]. Blocking NGF and its receptors, or the involved pathways, alongside chemotherapy could sensitize cancer cells to chemotherapy, preventing the development of drug resistance and possibly helping overcome or delay the emergence of drug resistance by targeting multiple pathways simultaneously [262,263]. Furthermore, inhibiting NGF or its receptors might enhance the efficacy of immunotherapies by reducing immunosuppression within the TME and preventing T cell dysfunction. The role of NGF and its receptor inhibitors currently seems limited in reducing symptomatology (mostly neuropathy) associated with chemotherapy [264].

The NTRK gene family encompasses three distinct members, namely NTRK1, NTRK2, and NTRK3, each responsible for producing TrkA, TrkB, and TrkC proteins. Rearrangement of these genes can occur either within the same chromosome or between different chromosomes, resulting in the fusion of two genes. This fusion event has the potential to create abnormal Trk proteins, ultimately promoting uncontrolled cellular proliferation [265]. The fusion of Trk genes gives rise to hybrid proteins that retain their functional tyrosine kinase domains, causing a sustained activation of downstream signaling pathways. This phenomenon is implicated in the development of various cancer types. Currently, there is ongoing research into inhibitors that exhibit promising tolerability and effectiveness, being evaluated in specific clinical trials across a range of cancer cell types.

To avoid NGF-related side effects, researchers have explored methods to stimulate the synthesis and release of endogenous NGF near damaged tissue, offering a potential avenue for therapeutic intervention.

## 7. Discussion

In this narrative review, we comprehensively explored the intricate molecular mechanisms through which inflammation and NGF impact tumor cell growth, survival, and death. We delved into the signaling pathways activated by inflammation and NGF, including the role in the microenvironment, interaction with the immune system, and the various molecules that play a role in tumorigenesis. NGF and inflammation have many meeting points and often act in a unanimous way to induce cancer. On the other hand, NGF is a complex molecule that has a pro-inflammatory and pro-tumoral role but also a negative feedback able to inhibit excessive inflammatory responses and tumor progression. The role of NGF is determined by the specific context and by the high and low expression of its specific receptors by the cells. By understanding the downstream effectors and cross-talk with other signaling cascades, we can gain deeper insights into the regulation of tumor cell proliferation and survival by NGF. Furthermore, we shed light on the multifaceted role of NGF in promoting tumor angiogenesis, its influence on EMT, and its ability to modulate the immune response within the TME. The reciprocal interactions between cancer cells

and the surrounding stromal and immune cells mediated by NGF highlight its potential as a promising target for cancer therapeutics. However, the paradoxical effects of NGF in cancer extend beyond its oncogenic potential, as accumulating evidence suggests its involvement in promoting tumor cell death in specific cellular contexts. Understanding the factors that drive this pro-apoptotic activity will be crucial for the development of novel therapeutic approaches that harness NGF-induced cell death pathways to combat cancer progression. Indeed, NGF plays a multifaceted role in human health, ranging from its essential functions in the nervous system to its potential implications in promoting cell proliferation and survival in cancer cells. Further research is needed to fully understand the extent of NGF's involvement in tumors and its potential therapeutic applications, especially in light of its role as a peculiar factor of the inflammatory process.

## 8. Conclusions

In conclusion, this report may provide a comprehensive overview of the multifaceted role of NGF as a component of the inflammatory phenomena in tumor cell growth and death, offering new perspectives on its potential as a therapeutic target in cancer treatment. The insights gained from unraveling the intricate interactions of NGF, inflammatory molecules, and the TME may pave the way for the development of precision therapies, enabling the exploitation of its dual nature for more effective and personalized cancer interventions.

**Author Contributions:** Conceptualization, S.T., G.F. and M.F.; methodology, S.T., G.F., P.T., M.L., F.F., B.C., L.C., A.P. and M.F.; validation, L.T., F.F., S.A. and A.G.; formal analysis, S.T., G.F. and M.F.; resources, S.T., G.F. and M.F.; data curation, L.T., F.F., S.A., B.C., P.T. and A.G.; writing—original draft preparation, S.T.; writing—review and editing, S.T., G.F. and M.F.; supervision, S.T., G.F., P.T., M.L., F.F., A.P., M.F., L.T., L.C., B.C. and A.G. All authors have read and agreed to the published version of the manuscript.

**Funding:** This research received no external funding.

**Institutional Review Board Statement:** Not applicable.

**Informed Consent Statement:** Not applicable.

**Data Availability Statement:** Not applicable.

**Acknowledgments:** The authors thank Sapienza Università di Roma, Italy, and IBBC-CNR of Rome, Italy.

**Conflicts of Interest:** The authors declare no conflicts of interest.

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
