# Peer review of "Nerve Growth Factor and the Role of Inflammation in Tumor Development"

_cimb, doi:10.3390/cimb46020062_

Round 1
Reviewer 1 Report
Comments and Suggestions for Authors
English language use is grammatically and idiomatically correct. The subject of NGF is an important one in oncology, psychiatry and neurology. A review of NGF in malignancy like this will be quite welcome. The role of NGF in promoting malignant growth has been underappreciated.
However, this paper requires better organization. It is obvious that the lead author has not read over the entire manuscript to edit and harmonize the various contributions, eliminate repetitions, delete empty or ambiguous phrases, once an abbreviation has been defined, use that abbreviation, etc. I would recommend the three papers below as examples of orderly introduction and presentation of NGF studies from which the authors might benefit from emulating. The prose is turgid. It can be considerably tightened without losing any information.
I think the authors might have attempted too much here ? They hop over NGF in TME, in tumor growth, in tumor prevention, in immune stimulation, in immune inhibition, in its epigenetic control, in angiogenesis, in stem cell function, in endothelium function, in EMT, etc. Recounting a litany of effects without the physiologic details - which in the case of NGF we know something about- will not help readers better understand NGF.
Since your subject is NGF in cancer, the paper must start, after a few paragraphs of Introduction, with a systematic overview of pro-NGF, NGF, TrkA, p75NTR, neuropilin-1. Also the proteolytic processing pro-NGF to mature NGF must be discussed as this has particular relevance in oncology and paths to treat. Only then systematically review evidence for this signaling system’s role in malignant growth. As this paper stands it launches into this role without giving readers an overview of what this system is and how it functions in normal physiology.
NGF as tumor suppressor and NGF as tumor growth facilitator must be reorganized into separate and specifically delineated sections.
Language use is unnecessarily florid or literary. Medical communication is different from literary communication. In medicine we want to say as much as possible with as few words as possible and omit generalities,
An example of such bad language use is line 100 “The presence of NGF and NGF receptors in cancer cells raised questions about NGF's involvement in promoting cell proliferation and cancer cell survival.” What does this say ?
Another example:
Line 241 “Interestingly, NGF has been implicated in the regulation of the tumor microenvironment and the immune response, and it may play a role in tumor immune escape in certain contexts.” What are the authors saying here ? That NGF has something to do with TME and something to do with immune escape. Why be coy ? State directly what you mean
I view lines 34 to 37 as formulaic, silly and self-evident.
Why not start your paper on line 41, “Understanding the relationship between inflammation and cancer has led…” Or better yet start with a careful overview of your subject, NGF/TrkA/neuropilin-1, then limit discussion of NGF in malignancy ?
The authors are not clear about the distinction between inflammation and immune systems. Neither are immunologists or the rest of us. The two interact, one can engage the other, and either can act without the other, albeit less efficiently if the two systems don’t coordinate. Rather than speak unclearly about this distinction/interaction, I suggest leaving this subject out. A similar objection to the diversion to subject of neurotrophins. Drop these.
It is not reasonable to discuss NGF/TrkA without discussing neuropilin-1, and pro-NGF and the proteolytic NGF maturation process.
I recognize that many papers like this one are appearing as it is now written, in the less prestigious journals. But I doubt this bland, discursive, poorly organized narrative would be of help to anyone. The subject is quite important. Teaching us what is known about NGF in cancer will be quite useful. I encourage the authors to give us such an overview.
==============================================
Baldassarro VA, Cescatti M, Rocco ML, Aloe L, Lorenzini L, Giardino L, Calzà L. Nerve growth factor promotes differentiation and protects the oligodendrocyte precursor cells from in vitro hypoxia/ischemia. Front Neurosci. 2023;17:1111170. doi:10.3389/fnins.2023.1111170.
Bradshaw RA, Pundavela J, Biarc J, Chalkley RJ, Burlingame AL, Hondermarck H. NGF and ProNGF: Regulation of neuronal and neoplastic responses through receptor signaling. Adv Biol Regul. 2015;58:16-27. doi: 10.1016/j.jbior.2014.11.003.
Cuello AC, Pentz R, Hall H. The Brain NGF Metabolic Pathway in Health and in Alzheimer's Pathology. Front Neurosci. 2019;13:62. doi: 10.3389/fnins.2019.00062.
Author Response
Answers to the criticisms raised by Reviewer 1
English language use is grammatically and idiomatically correct. The subject of NGF is an important one in oncology, psychiatry and neurology. A review of NGF in malignancy like this will be quite welcome. The role of NGF in promoting malignant growth has been underappreciated.
Reply: we thank the reviewer for the positive comments
However, this paper requires better organization. It is obvious that the lead author has not read over the entire manuscript to edit and harmonize the various contributions, eliminate repetitions, delete empty or ambiguous phrases, once an abbreviation has been defined, use that abbreviation, etc. I would recommend the three papers below as examples of orderly introduction and presentation of NGF studies from which the authors might benefit from emulating. The prose is turgid. It can be considerably tightened without losing any information. I think the authors might have attempted too much here ? They hop over NGF in TME, in tumor growth, in tumor prevention, in immune stimulation, in immune inhibition, in its epigenetic control, in angiogenesis, in stem cell function, in endothelium function, in EMT, etc. Recounting a litany of effects without the physiologic details - which in the case of NGF we know something about- will not help readers better understand NGF.
Reply: according to the comments of the two reviewers, the paper has been revised to harmonize the various contributions, eliminate repetitions, delete empty or ambiguous phrases and once an abbreviation has been defined, use that abbreviation. All the changes in the paper have been highlighted in light yellow.
Since your subject is NGF in cancer, the paper must start, after a few paragraphs of Introduction, with a systematic overview of pro-NGF, NGF, TrkA, p75NTR, neuropilin-1. Also, the proteolytic processing pro-NGF to mature NGF must be discussed as this has particular relevance in oncology and paths to treat. Only then systematically review evidence for this signaling system’s role in malignant growth. As this paper stands it launches into this role without giving readers an overview of what this system is and how it functions in normal physiology.
Reply: as suggested, we included new paragraphs including new info on pro-NGF, NGF, TrkA, p75NTR, and neuropilin-1. Furthermore, the issue regarding the processing pro-NGF to mature NGF has been discussed (pages 2 and 3, new info in Table 1 of the revised paper).
NGF as tumor suppressor and NGF as tumor growth facilitator must be reorganized into separate and specifically delineated sections.
Reply: as suggested, a section dealing with this issue has been included in the revised paper (page 4 of the revised paper).
Language use is unnecessarily florid or literary. Medical communication is different from literary communication. In medicine we want to say as much as possible with as few words as possible and omit generalities,
An example of such bad language use is line 100 “The presence of NGF and NGF receptors in cancer cells raised questions about NGF's involvement in promoting cell proliferation and cancer cell survival.” What does this say ?
Reply: as requested, the sentence has been revised.
Another example:
Line 241 “Interestingly, NGF has been implicated in the regulation of the tumor microenvironment and the immune response, and it may play a role in tumor immune escape in certain contexts.” What are the authors saying here ? That NGF has something to do with TME and something to do with immune escape. Why be coy ? State directly what you mean
Reply: as requested, the sentence has been deleted.
I view lines 34 to 37 as formulaic, silly and self-evident. Why not start your paper on line 41, “Understanding the relationship between inflammation and cancer has led…” Or better yet start with a careful overview of your subject, NGF/TrkA/neuropilin-1, then limit discussion of NGF in malignancy ?
Reply: We thank the reviewer for the suggestion. According to this, we revised the sentences.
The authors are not clear about the distinction between inflammation and immune systems. Neither are immunologists or the rest of us. The two interact, one can engage the other, and either can act without the other, albeit less efficiently if the two systems don’t coordinate. Rather than speak unclearly about this distinction/interaction, I suggest leaving this subject out. A similar objection to the diversion to subject of neurotrophins. Drop these.
Reply: according to the comments of the two reviewers the section dealing with inflammation has been rewritten to better harmonize the process of carcinogenesis
It is not reasonable to discuss NGF/TrkA without discussing neuropilin-1, and pro-NGF and the proteolytic NGF maturation process.
Reply: as requested, further info on the correlation between NGF/TrkA and neuropilin-1, pro-NGF, and the proteolytic NGF maturation processes have been included in the revised paper (pages 2 and 3; new info on Table 1).
I recognize that many papers like this one are appearing as it is now written, in the less prestigious journals. But I doubt this bland, discursive, poorly organized narrative would be of help to anyone. The subject is quite important. Teaching us what is known about NGF in cancer will be quite useful. I encourage the authors to give us such an overview.
Reply: we want to stress the point that this work is designed to correlate the events connected with inflammatory processes and the onset of certain types of tumors according also to our long-lasting experience in the field of NGF and related diseases.
==============================================
Reviewer 2 Report
Comments and Suggestions for Authors
The author reviewed the role of the nerve Growth Factor and Inflammation in Tumor Development. This review gives comprehensive knowledge of NGF in tumor, while some concerns should be resolved:
1. Although this manuscript is well prepared, the language should be polished by a native speaker at deep extent.
2. Abbreviate of phrases should be explained when they were used for the first time in the manuscript.
3. The title should be modified, when you use inflammation, you should also introduce inflammation related other situation such as hepatitis.
4. Materials and Methods could be removed in a review.
5. The author should first give brief introduction of NGF.
6. Some parts of the paper should be organized. For example, in part 3, “tumor growth” part belongs to part 4, “tumor” part.
7. The introduction includes cancer resistance, but in the paper the chemoresistance related content is scare; role of NGF in drug resistance should be discussed, either in the growth of the inherently resistant population or induction/inhibition of initially drug sensitive cells.
8. The role of NGF in tumor microenvironments was mentioned, while the detailed mechanism should also be introduced, for example, whether they could reduce or induce T cell dysfunction, whether NGF correlated with immunotherapy resistance.
9. Nerve growth factor regulates liver cancer cell polarity and motility, which should be briefly summarized in the reviewed.
10. Most reference should be published in the latest 5 years; some great reviews should be referred.
Comments on the Quality of English LanguageMinor editing needed.
Author Response
Answers to the criticisms raised by Reviewer 2
The author reviewed the role of the nerve Growth Factor and Inflammation in Tumor Development. This review gives comprehensive knowledge of NGF in tumor, while some concerns should be resolved:
- Although this manuscript is well prepared, the language should be polished by a native speaker at deep extent.
Reply: as suggested, we improved the English quality of the paper.
- Abbreviate of phrases should be explained when they were used for the first time in the manuscript.
Reply: as requested, we defined the abbreviations throughout the manuscript.
- The title should be modified, when you use inflammation, you should also introduce inflammation related other situation such as hepatitis.
Reply: We modified the title and added a new chapter (2.2, pages 3 and 4 of the revised text, text highlighted in light yellow) introducing inflammation and its role in various conditions including hepatitis.
- Materials and Methods could be removed in a review.
Reply: The section on materials and methods has been removed
- The author should first give brief introduction of NGF.
Reply: We added a new chapter (2.1, pages 2 and 3 of the revised text, text highlighted in light yellow) for introducing NGF.
- Some parts of the paper should be organized. For example, in part 3, “tumor growth” part belongs to part 4, “tumor” part.
Reply: we rearranged the chapters as requested.
- The introduction includes cancer resistance, but in the paper the chemoresistance related content is scarse; role of NGF in drug resistance should be discussed, either in the growth of the inherently resistant population or induction/inhibition of initially drug sensitive cells.
Reply: we introduced the role of NGF in the case of chemoresistance and further discussed it in chapter 6 (pages 13 and 14, of the revised text, text highlighted in light yellow).
- The role of NGF in tumor microenvironments was mentioned, while the detailed mechanism should also be introduced, for example, whether they could reduce or induce T cell dysfunction, whether NGF correlated with immunotherapy resistance.
Reply: we extended this concept through the text and modified and updated Section 3 on tumor immunoescape (text highlighted in light yellow).
- Nerve growth factor regulates liver cancer cell polarity and motility, which should be briefly summarized in the reviewed.
Reply: done (page 4, lines 184-190 of the revised text). Furthermore, we better introduced the role of NGF in liver cancer in Table 1.
- Most reference should be published in the latest 5 years; some great reviews should be referred.
Reply: as suggested, we updated the references
Round 2
Reviewer 1 Report
Comments and Suggestions for Authors
This is a problematic paper. I cannot provide a detailed response quickly. The authors require help from a colleague to limit and tighten their wandering recount. I appreciate the tremendous work the authors have dedicated to this paper and I want to have this potentially valuable paper published but as it stands no one will read it. Authors - please get help. They must restrict their focus and make their report clearer. Would the authors consider starting their paper on line 80 ? I cannot see how this paper can be corrected without restriction of focus. Can the authors revise their figure 1 ? The listed elements are just floating around macrophage and lymphocyte. Can the authors restrict their paper to NGF ? Remember this is a didactic paper - the authors wish to report to us how NGF is made and how it helps tumors grow. I object to the term "brain tumors" line 85. CNS lymphoma, a melanoma metastasis, ependomas, and grade 2 gliomas are all "brain tumors but they do not share any particular relationship with each other with respect to NGF do they ? re. a comment in Table 1 re. gastric cancer "NGF Trk receptors may play a key role in cell apoptosis by a Ras or Raf signal transduction pathway." is an example among dozens of others in this paper of the authors being vague or coy. "play a role" means inhibit, or stimulate, delay, hasten, be a requirement for, etc. line 344 breaks all rules for medical reporting: "Notably, a drug approved by the Food and Drug Administration (FDA) initially designed for migraine treatment proves effective in this regard [127]." Then they don't further discuss this. The authors state NGF can promote or inhibit cancer growth. The fault could be my own, but I don't see the growth inhibiting aspects of NGF as significant. If the authors do see such growth inhibiting data they must explan that more clearly. There are now several minor typos that require correcting. A careful exposition of TykA and TykB physiology is required with delineation of how these contribute to malignant growth. Also - and agin the fault could be mine - if the authors see how TrkA and TrkB contribute to malignant growth suppression they must make that clearer because I don't see it. Comments on the Quality of English Language
This is a problematic paper. I cannot provide a detailed response quickly. The authors require help from a colleague to limit and tighten their wandering recount. I appreciate the tremendous work the authors have dedicated to this paper and I want to have this potentially valuable paper published but as it stands no one will read it. Authors - please get help. They must restrict their focus and make their report clearer. Would the authors consider starting their paper on line 80 ? I cannot see how this paper can be corrected without restriction of focus. Can the authors revise their figure 1 ? The listed elements are just floating around macrophage and lymphocyte. Can the authors restrict their paper to NGF ? Remember this is a didactic paper - the authors wish to report to us how NGF is made and how it helps tumors grow. I object to the term "brain tumors" line 85. CNS lymphoma, a melanoma metastasis, ependomas, and grade 2 gliomas are all "brain tumors but they do not share any particular relationship with each other with respect to NGF do they ? re. a comment in Table 1 re. gastric cancer "NGF Trk receptors may play a key role in cell apoptosis by a Ras or Raf signal transduction pathway." is an example among dozens of others in this paper of the authors being vague or coy. "play a role" means inhibit, or stimulate, delay, hasten, be a requirement for, etc. line 344 breaks all rules for medical reporting: "Notably, a drug approved by the Food and Drug Administration (FDA) initially designed for migraine treatment proves effective in this regard [127]." Then they don't further discuss this. The authors state NGF can promote or inhibit cancer growth. The fault could be my own, but I don't see the growth inhibiting aspects of NGF as significant. If the authors do see such growth inhibiting data they must explan that more clearly. There are now several minor typos that require correcting. A careful exposition of TykA and TykB physiology is required with delineation of how these contribute to malignant growth. Also - and agin the fault could be mine - if the authors see how TrkA and TrkB contribute to malignant growth suppression they must make that clearer because I don't see it.
Author Response
Answers to the criticisms raised by Reviewer 1
This is a problematic paper. I cannot provide a detailed response quickly. The authors require help from a colleague to limit and tighten their wandering recount. I appreciate the tremendous work the authors have dedicated to this paper and I want to have this potentially valuable paper published but as it stands no one will read it. Authors - please get help.
Reply: It’s crystal clear that we and the reviewer have different opinions. However, according to your quite helpful comments, we made further changes in the paper.
They must restrict their focus and make their report clearer. Would the authors consider starting their paper on line 80 ?
Reply: this paper was aimed for the Journal Current Issues in Molecular Biology, for the Section “Molecular Medicine” and specifically for the Special Issue “Advances in Understanding Molecular Basis of Inflammatory Diseases”. We tried to combine info from different fields of research, namely, NGF, inflammation and cancer. We do believe that this combination of factors is the real novelty of a report like this. The short intro is supposed to associate these 3 lines of investigation.
Can the authors revise their figure 1 ? The listed elements are just floating around macrophage and lymphocyte.
Reply: as requested, Figure 1 was deeply revised and updated
Can the authors restrict their paper to NGF ? Remember this is a didactic paper - the authors wish to report to us how NGF is made and how it helps tumors grow.
Reply: as stated before, we aimed to combine info from different fields of research, namely, NGF, inflammation and cancer. We do strongly believe that this combination of factors is the real novelty of a report like this.
I object to the term "brain tumors" line 85. CNS lymphoma, a melanoma metastasis, ependomas, and grade 2 gliomas are all "brain tumors but they do not share any particular relationship with each other with respect to NGF do they ?
Reply: we thank the reviewer for the suggestion. We updated Table 1 to include only tumors sharing similarities (pages 9, 10 and 11, text highlighted in light yellow).
- a comment in Table 1 re. gastric cancer "NGF Trk receptors may play a key role in cell apoptosis by a Ras or Raf signal transduction pathway." is an example among dozens of others in this paper of the authors being vague or coy. "play a role" means inhibit, or stimulate, delay, hasten, be a requirement for, etc.
Reply: again, we thank the reviewer for the suggestion. We updated Table 1 (pages 9, 10 and 11, text highlighted in light yellow).
line 344 breaks all rules for medical reporting: "Notably, a drug approved by the Food and Drug Administration (FDA) initially designed for migraine treatment proves effective in this regard [127]." Then they don't further discuss this. The authors state NGF can promote or inhibit cancer growth. The fault could be my own, but I don't see the growth inhibiting aspects of NGF as significant. If the authors do see such growth inhibiting data they must explan that more clearly.
Reply: we sincerely apologize for the typing mistakes. The sentence was removed.
There are now several minor typos that require correcting.
Reply: as suggested, the paper will undergo careful English editing by the dedicated MDPI service.
A careful exposition of TykA and TykB physiology is required with delineation of how these contribute to malignant growth. Also - and agin the fault could be mine - if the authors see how TrkA and TrkB contribute to malignant growth suppression they must make that clearer because I don't see it.
Reply: as suggested, updated info on TrkA and TrkB physiology in malignant growth was added (page 12, lines 424-443, text highlighted in light yellow).
Reviewer 2 Report
Comments and Suggestions for Authors
All concerns resolved.
Author Response
We thank the reviewer for the positive comment.